# Ethical Dilemmas and Legal Responsibilities in Patient Care: An Analysis of Hospital Safety

**DOI:** 10.3390/healthcare13212800

**Published:** 2025-11-04

**Authors:** Andrada-Georgiana Nacu, Dan-Alexandru Constantin, Liliana Marcela Rogozea

**Affiliations:** Department of Fundamental, Prophylactic and Clinical Sciences, Faculty of Medicine, Transilvania University of Brasov, 500019 Brasov, Romania; englober@icloud.com (D.-A.C.); r_liliana@yahoo.com (L.M.R.)

**Keywords:** hospital ethics, informed consent, medical error disclosure, AI, institutional responsibility, end-of-life care, patient safety, healthcare law

## Abstract

**Background/Objectives:** This systematic review explores the evolving landscape of ethical dilemmas and legal responsibilities in hospital-based patient care, with particular attention to how they intersect with institutional safety. Drawing from 40 studies published within the last decade and supplemented by seminal theoretical works, the review examines clinical, technological, legal, and policy-driven challenges that compromise ethical clarity and organisational accountability. **Methods:** Methods included a structured literature search across major academic databases, guided by inclusion and exclusion criteria aligned with PRISMA guidelines and a quality appraisal ensuring inclusion of only medium- and high-quality studies. **Results:** Findings reveal that ethical complexity has shifted from individual decision-making to system-level vulnerabilities, particularly in contexts involving artificial intelligence (AI), data governance, consent procedures, and end-of-life care. Moreover, hospitals often lack sufficient protocols for disclosure, cross-border telemedicine accountability, and ethically responsive infrastructure. The results support a growing call for ethics-by-design approaches, where ethical reflexivity is embedded into institutional processes and digital systems. **Conclusions:** Overall, ethical resilience in hospital care depends not only on clinical training but on proactive organisational structures that support transparency, equity, and patient dignity.

## 1. Introduction

The delivery of safe and ethical patient care within hospitals sits at a persistent intersection of clinical practice, legal mandates, and evolving ethical frameworks. Clinicians routinely navigate situations that require balancing professional obligations with patients’ rights. These decisions unfold within, and are constrained by, institutional policies. These dilemmas—often involving informed consent, privacy, autonomy, and prioritisation of care—are further complicated by shifting legal responsibilities around medical liability, malpractice, and data governance [1,2]. In recent years, heightened global attention to hospital safety has been driven by preventable harm and medical error; a 2025 study underscores the risks of unregulated clinical decision—making and the need for robust ethical–legal scaffolding in acute settings [3]. In parallel, the expansion of telemedicine and AI-enabled care introduces new complexities, including algorithmic bias, partial delegation of clinical judgement to machines, and jurisdictional ambiguities in virtual practice [4,5,6].

Ethical principles of beneficence, non-maleficence, justice, and respect for autonomy remain foundational to medical practice [7,8]. Yet their application becomes contentious under time pressure, resource constraints, or public health exigencies. Disagreements arise over the limits of patient autonomy and the role of professional discretion, as well as the ethical permissibility of withholding interventions in cases judged medically futile [9,10]. Layered onto these debates is a dynamic legal environment: regulations, institutional policies, and rights instruments increasingly shape bedside practice. Contemporary regulatory tools—such as the EU AI Act, the FDA’s Good Machine Learning Practice and Predetermined Change Control Plans, and WHO guidance on AI (AI) for health—signal a shift from general principles to enforceable duties concerning risk management, transparency, logging, and human oversight [11,12,13,14,15,16,17].

Hospitals warrant special focus because they are the highest-stakes nodes of healthcare: technologically dense, tightly regulated, and characterised by rapid, team-based decisions under uncertainty. Ethical and legal tensions are amplified in this setting, where duties of disclosure and safety operate at the institutional as well as the individual level. Safety science shows that outcomes reflect system design as much as bedside judgement [18], and leading consent jurisprudence (Montgomery v Lanarkshire Health Board in the UK and Canterbury v Spence in the U.S.) grounds disclosure and patient involvement in enforceable duties that hospitals must operationalise through policy, documentation, and training [19,20].

Against this backdrop, the present review synthesises contemporary scholarship at the intersection of ethical dilemmas, legal responsibilities, and hospital safety. Drawing on diverse contexts—emergency response, AI integration, cross-border and virtual care, and end-of-life practices—we identify converging themes and unresolved controversies, situating them within emerging regulatory and governance landscapes. Our objective is to map best practices, surface ethical blind spots, and develop recommendations that strengthen a culture of ethical compliance and patient-centred care while aligning institutional routines with applicable legal and regulatory requirements.

Beyond the hospital microsystem, governance capacity is also shaped by regional innovation networks and ownership structures, while workforce capability remains a prerequisite for safe digital transformation. Recent evidence shows (i) networks in the advanced medical-equipment industry—driven by economic development, technological capability, and policy support—structure regional digital-health capacity and collaboration [21]; (ii) workforce preparation matters—new-media pharmacology teaching outperforms traditional approaches among nursing/junior-college students [22]; and (iii) ownership models correlate with outcomes and costs—a multi-hospital study in China found higher in-hospital pneumonia mortality and higher expenses in private hospitals than in public facilities [23]. These insights motivate our focus on organisational accountability and capability within an ethical–legal frame.

## 2. Materials and Methods

This systematic review was conducted in accordance with the Preferred Reporting Items for Systematic Reviews and Meta-Analyses (PRISMA) 2020 guidelines to ensure methodological transparency, reproducibility, and scientific rigour. No protocol was registered in PROSPERO, as the review includes interdisciplinary elements (ethics, law, policy) beyond PROSPERO’s registration scope.

We adopted an integrative evidence synthesis tailored to a question that spans ethics, law, and organisational governance in hospitals. PRISMA-2020 was used to transparently report the identification, screening, and inclusion of peer-reviewed empirical and review studies. In parallel, because the topic explicitly concerns legal duties and enforceable governance, we conducted a targeted scan of authoritative normative sources (statutes, case law, regulatory guidance, and WHO/EU/FDA frameworks). Excluding such sources would have biased the review away from the very obligations hospitals must meet. Accordingly, our corpus comprises (i) an empirical stream appraised and counted under PRISMA and (ii) a normative stream used to contextualise and operationalise the empirical findings into legally actionable recommendations. The two streams were synthesised qualitatively but were kept analytically distinct for selection and appraisal.

Eligibility criteria:Empirical/review stream (PRISMA): peer-reviewed primary studies and reviews addressing ethical dilemmas and/or legal responsibilities in hospital care and their implications for patient safety (2015–2025).Normative stream (policy/legal): statutes, binding regulations, official regulatory guidance, and high-authority policy frameworks directly specifying obligations for hospitals/clinicians/developers relevant to the review question (no date restriction for seminal case law; current versions for living guidance). Normative sources were not treated as empirical evidence but as governance anchors for interpreting and implementing the empirical themes.

Five bibliographic databases (PubMed, Scopus, Web of Science, ScienceDirect, Google Scholar) were searched for English-language records (2015–April 2025). Search strings combined MeSH and free-text terms around ethical dilemmas, legal responsibility/liability, patient safety, and hospital. In a parallel policy scan, we performed structured website searches on authoritative domains (e.g., eur-lex.europa.eu, ec.europa.eu for the EU AI Act; fda.gov for GMLP and PCCP; who.int for WHO AI ethics/LMM guidance) and citation-chased to landmark case law (e.g., Montgomery; Canterbury). This two-track strategy reflects the rapid and jurisdiction-specific evolution of governance in digital health and patient safety.

The final search was completed on 1 April 2025. Two reviewers independently screened titles/abstracts and full texts, with disagreements resolved by discussion. Data extraction covered study design, setting, ethical–legal focus, and main findings. We applied predefined quality-appraisal criteria to include only medium- and high-quality studies; normative sources were not appraised as empirical evidence but used to contextualise and operationalise findings.

The literature search yielded a total of 164 records across multiple databases. Specifically, 68 articles were identified in PubMed, 42 in Scopus, 31 in Web of Science, 18 in ScienceDirect, and 23 relevant records from Google Scholar after manual screening of the first 200 results. Additionally, six records were retrieved from the WHO IRIS repository, and eight grey literature sources were identified via ProQuest Dissertations, ResearchGate, and NHS Library. After 48 duplicates were removed, a total of 116 unique records remained for title and abstract screening. Of the 116 screened articles, 59 were excluded for not meeting the inclusion criteria. The remaining 57 full-text articles underwent a detailed eligibility assessment, after which 40 studies were deemed eligible for inclusion in the final synthesis. The study selection process is depicted in the PRISMA flow diagram (Figure 1).

## 3. Results

To integrate the ethical, legal and institutional strands identified above, we propose a concise Triadic Governance Model that aligns three domains—ethical, legal, and organizational—and links inputs (risk/data/algorithms) to process controls (oversight, logging, audit) and outcomes (transparency, equity, dignity, safety). The legend maps (Figure 2) each control to its legal anchor (e.g., EU AI Act articles on human oversight and logging; FDA GMLP/PCCP; Montgomery/Canterbury; GDPR) and to a concrete institutional mechanism (e.g., ethics committee, CIRS-Ethics, and DPO-led data release). Table 1 presents a summary of the included sources (theme, context, study type, theoretical lens, and quality).

We categorised the studies into three cross-cutting domains (Table 2)—ethical dimensions, legal implications, and institutional mechanisms. These thematic domains often intersect, especially in discussions of AI that involve both privacy and clinical decision-making responsibilities. The 40 studies encompass various study types, revealing an academic inclination toward theoretical exploration over empirical investigation; the distribution of study types is shown in Table 3.

To avoid misclassification, we separate observational/quantitative from qualitative.

The corpus is dominated by conceptual/policy/guidance work (82.5%) with a smaller empirical base (17.5%), which we factored into how heavily each result weighed in the synthesis (high-quality evidence prioritised).

### 3.1. Ethical Dimensions

Clinical practice within hospital systems is often marked by ethical ambiguity, especially when medical actions intersect with legal frameworks and evolving technologies.

Ethical uncertainty in hospitals increasingly originates in system conditions rather than from a lack of principles. Time pressure, team-based work, digitally mediated information, and fragmented responsibility create situations in which clinicians can neither fully explain nor fully control the reasons behind a course of action. In this environment, person-centredness functions less as a communication ideal and more as a design criterion for everyday practice: care remains ethically sound only when workflows enable clinicians to recognise patients as persons with situated values and capabilities, and to justify decisions in terms the patient can grasp [44].

What renders these situations ethically fragile is not merely cognitive load but opacity about how decisions are produced and who is answerable for them. When data pipelines, triage rules, or decision support remain obscure at the bedside, clinicians struggle to provide the kind of reason-giving that underpins informed choice and trust. Disclosure after harm illustrates the same logic from another angle: transparency is an ethical infrastructure rather than an after-the-fact virtue, and organisations that normalise open discussion show stronger cultures of safety and learning [44,45].

Throughout this review we therefore treat ethical performance as a property of systems: explainability, fair data practices, and routinised deliberation are preconditions for defensible care, not optional refinements. The legal allocation of duties that makes such conditions enforceable is analysed in Section 3.2, while the institutional routines that convert principles into practice (e.g., audit trails, ethics rounds, data-release oversight) are detailed in Section 3.3.

Data governance in acute care and emergency settings presents another ethical minefield. References [1,3] reveal that in high-pressure clinical environments, consent to collect and share data is often assumed rather than granted. This undermines both autonomy and transparency, especially when data are later repurposed for research, training, or algorithm development without the patient’s knowledge. The authors advocate for “procedural ethics” embedded at the systems level, ensuring that ethical compliance is as automated and routine as the data processes it accompanies. Telemedicine, increasingly normalised in post-pandemic healthcare, adds further complexity. Wahyudin et al. [4] examine legal ambiguity in cross-border teleconsultations, finding that practitioners often operate without clear legal guidance on jurisdiction, licensure, or malpractice standards. The ethical implications extend beyond legality: patients may not fully understand their remote provider’s accountability or the limitations of virtual care. This opacity compromises both trust and autonomy, rendering consent practically symbolic in some settings.

The increasing integration of digital technologies, particularly AI and machine learning (ML), has shifted the ethical landscape in hospital-based care. Ethical complexity is no longer confined to interpersonal relationships between physicians and patients; it now includes technical architectures, opaque algorithms, data pipelines, and system-wide automation. This section explores how technologies not only support, but reshape, ethical decision-making and how hospitals must adapt their frameworks to manage emerging risks and responsibilities. Roy et al. [7] provide a foundational overview of AI and ML in clinical environments, arguing that such systems function within ethical regimes that remain underdeveloped compared to their technological sophistication. Their review highlights the contradiction at the heart of AI deployment: while AI is introduced to reduce variability and improve precision, it simultaneously introduces ethical variability due to lack of transparency and accountability mechanisms. Roy and colleagues advocate for “ethics-by-design,” a paradigm in which AI development includes preemptive ethical modelling, testing for bias, and ensuring human interpretability.

Singh and Rabinstein [8] echo this sentiment in the domain of acute neurology, where algorithms are increasingly used to triage stroke patients or predict seizures. However, such tools can amplify ethical challenges by prioritising efficiency over person-centredness (i.e., treating patients as persons with situated values and capabilities). Their study notes that clinicians often receive AI-based recommendations without being able to interrogate the data source, training logic, or probabilistic assumptions. In time-sensitive contexts, physicians must act on decisions they cannot fully understand—a form of delegated agency that creates friction with traditional models of professional responsibility.

The matter becomes further complicated in the context of caregiving, as Fox [9] explores in his dissertation on family-member caregivers in U.S. hospital systems. As care delivery becomes technologically mediated through apps, monitoring systems, and digital communications, ethical questions emerge about surrogate decision-making, data transparency, and caregiver burden. The institutional tendency to equate technological access with informed consent, Fox warns, creates blind spots where ethical violations may go unrecognised until harm occurs. Hospitals must therefore reconsider how ethical support structures extend beyond professional clinicians to encompass non-clinical actors embedded in digital care networks. From an institutional perspective, Stark et al. [10] offer empirical insights on how health data reuse creates legal and ethical tension, especially when clinicians are both data providers and beneficiaries. Based on interviews with German healthcare professionals, the authors reveal that few staff members are trained to understand the implications of data analytics on patient rights. Even when anonymisation protocols are observed, ethical discomfort remains regarding non-consensual secondary use. This indicates that ethical frameworks must evolve beyond the binary of consent/no-consent to address systemic reuse in learning health systems.

Roy et al. [7] delve into the ramifications of AI integration in clinical care. Their review suggests that while AI may reduce bias in certain areas, it introduces new forms of ethical opacity. For example, machine learning models often lack explainability, and clinicians may find themselves enacting decisions they do not fully understand. This challenges the physician’s role as moral agent and potentially undermines the standard of informed decision-making. Roy et al. advocate for a shift from retrospective regulation to prospective ethical design—integrating ethical constraints during algorithm development rather than reacting to ethical failures after deployment. Singh and Rabinstein [8] echo these concerns from a neurology perspective, noting that in stroke units and neurological ICUs, decisions are often made within minutes under extreme pressure. The introduction of AI in such environments raises serious concerns: can clinicians realistically evaluate the moral weight of algorithm-based triage under extreme temporal constraints? The authors propose that ethical reflexivity must become part of routine training, not just ethics committee deliberation. Collectively, these studies depict a clinical ecosystem in which ethical clarity is the exception rather than the norm. Whether dealing with novel technologies, uncertain prognoses, or cross-border legal environments, healthcare professionals frequently operate in zones of ethical opacity. What unites these cases is not the absence of ethical concern, but the fragmentation of ethical authority between clinicians, patients, institutions, and, increasingly, machines. As such, future hospital ethics cannot remain focused solely on clinical bedside interactions [24]; it must evolve into a multilevel system that includes legal harmonisation, algorithmic accountability, and institutional governance.

To move from a “shared agency” problem statement to an actionable allocation of duties, we align with current regulatory instruments that distribute obligations across developers (“providers”), deployers (hospitals), and clinicians. In the EU, the AI Act classifies most clinical AI as high-risk and imposes specific duties before and during deployment: lifecycle risk management; data governance for training/validation/testing datasets; technical documentation and logging to ensure traceability; clear information for deployers; appropriate human oversight; and requirements for robustness, accuracy, and cybersecurity. Together, these requirements establish an auditable chain of responsibility between model developers and hospital deployers; importantly, Article 14 mandates human oversight that can prevent or minimise risks and enable meaningful intervention or override [11,41].

In the United States, the FDA’s Good Machine Learning Practice (GMLP) principles set expectations for data and model management across the lifecycle, and the final guidance on Predetermined Change Control Plans (PCCPs) allows sponsors to pre-specify which algorithm changes are permitted and how they will be validated and implemented. A PCCP must describe planned modifications, the associated development/validation methodology, and a safety-effectiveness impact assessment so that iterative model improvements can occur under agreed guardrails without losing accountability. In practice, this clarifies responsibilities across manufacturers, hospital deployers, and clinicians during post-market learning [13,14].

Complementary WHO guidance adds an institutional ethics layer: the 2021 framework articulates principles for transparency, accountability, bias mitigation and appropriate human oversight across the AI lifecycle, while the 2024–2025 guidance on large multimodal models (LMMs) sets safeguards for generative systems used in clinical or administrative workflows. Read together, these instruments provide concrete levers for assigning obligations across AI developers, hospitals, and clinical teams, answering the practical question of who is responsible when many parties (including AI) are involved [15,17].

These tensions grow even more acute in end-of-life scenarios, where patients may lack decisional capacity and surrogate decision-makers are absent or unprepared. Mulrooney et al. [5] analyse PEG (percutaneous endoscopic gastrostomy) placement in a patient with advanced Creutzfeldt–Jakob disease and dementia. The clinical team faced ethical paralysis: proceeding with the intervention risked violating non-maleficence, yet forgoing it conflicted with institutional protocols. In these circumstances, the principle of respect for autonomy becomes fractured, often reduced to previously documented preferences rather than real-time patient input. This case illustrates the profound ethical instability that can arise when biomedical technologies (like feeding tubes) outpace the moral clarity of their application. Turbanova and Salimova [6] shift focus to simulation-based training environments and their impact on ethical implications. Simulations are designed as educational tools, but the authors highlight their formative role in shaping clinicians’ ethical intuitions. If training scenarios oversimplify ethical dilemmas or present unrealistic resolutions, practitioners may carry skewed ethical expectations into real patient care. Thus, the fidelity of ethical challenges in simulation design becomes a matter of patient safety, not just pedagogical preference.

End-of-life care is one of the most ethically complex domains in hospital medicine. The questions it raises are deeply personal, often emotionally fraught, and legally nuanced [25]. What complicates these decisions further is that they often occur in contexts of clinical uncertainty, impaired patient capacity, and institutional policies that may not reflect the granular ethical needs of individual cases. This subsection explores how the reviewed literature frames ethical and legal responsibility in end-of-life contexts, particularly with regard to the principle of autonomy, the role of surrogates, and the boundaries of medical futility.

A compelling case analysis by Mulrooney et al. [5] examines the ethical deliberations surrounding PEG (percutaneous endoscopic gastrostomy) tube placement in a patient diagnosed with Creutzfeldt–Jakob disease and advanced dementia. The care team faced a dilemma: the patient lacked decision-making capacity, and family input was ambiguous. The institutional policy favoured intervention, while the clinical team leaned toward palliation. In such situations, autonomy becomes fractured, not in principle, but in application. The case demonstrates that end-of-life decisions cannot be made solely based on documented preferences or proxies; they require a holistic ethical judgment that considers prognosis, suffering, dignity, and values interpretation.

Clinical ethics texts such as those by Jonsen, Siegler, and Winslade [45] provide structured frameworks for such decision-making, using the “Four-Box Method” to weigh medical indications, patient preferences, quality of life, and contextual features. However, the utility of such models is challenged when patients have neurodegenerative conditions or fragmented family dynamics. Here, decision-making moves from normative ethics to narrative ethics, where the personal history of the patient and the values of those involved must be reconstructed to approximate a just choice. Ref. [26], in their guide for clinicians, emphasises the importance of distinguishing between “allowing to die” and “causing death.” They argue that institutional language often blurs this boundary, creating moral distress for clinicians tasked with implementing decisions they did not initiate. Hospitals that lack clear, actionable policies for end-of-life care inadvertently shift the burden of decision onto the shoulders of individual physicians, who may be untrained or unsupported in ethical deliberation. This not only compromises patient dignity but also exacerbates clinician burnout and moral injury.

Berlinger, Jennings, and Wolf [24,46] expand this framework by integrating ethics committee consultation into hospital policy. Their guidelines propose a structured process where palliative interventions, do-not-resuscitate (DNR) orders, and withdrawal of life-sustaining treatment (LST) are subject to multidisciplinary deliberation. The strength of this model lies in institutionalising ethics, moving decision-making from private negotiation to collective moral reasoning. However, its implementation is inconsistent across institutions, especially in systems with limited ethics infrastructure or under-resourced settings. The psychological and legal burden of medical error in end-of-life care is addressed by ref. [25], who examine how disclosure of adverse outcomes—especially during palliative treatment—interacts with litigation fears and communication breakdowns. Their findings reveal that failure to disclose does not reduce risk but instead increases dissatisfaction, moral distress, and reputational harm. The authors propose proactive communication strategies rooted in transparency, apology, and shared narrative, suggesting that ethical responsibility must include acknowledgement of failure, not just its avoidance.

Organisational and workforce pressures—such as pharmacy staffing gaps—compound the fragmentation of hospital care and constrain how patient autonomy can be enacted in practice [27]. Autonomy, often heralded as the foundational principle in bioethics, is increasingly difficult to uphold in the fragmented realities of hospital care. Beauchamp and Childress [28] define autonomy as both self-rule and freedom from controlling interference. Yet, in end-of-life contexts, autonomy is frequently relational, shaped by family interpretation, clinician framing, and institutional constraint. For example, when a patient’s prior expressed wishes conflict with current clinical realities, clinicians may find themselves torn between substituted judgment and best interest standards, neither of which fully resolves the ethical impasse.

In their influential article, Gillon [47] clarifies that the four principles of medical ethics—autonomy, beneficence, non-maleficence, and justice—must be contextualised, not ranked. In terminal care, beneficence and non-maleficence often become dominant, pushing autonomy into a procedural rather than substantive role. For instance, a decision to discontinue dialysis in a patient with multi-organ failure may be ethically justified even in the absence of prior explicit consent, provided it aligns with palliative goals and minimises suffering. Donabedian’s work on care quality [43] reminds us that ethical end-of-life care is inseparable from systemic quality assurance. Hospitals that fail to provide ethics training, palliative support, or interdisciplinary decision-making structures create conditions where ethically suboptimal care becomes normalised. Ref. [48] reinforce this view by noting that the safest systems are those that actively integrate ethical reflexivity into patient safety protocols. Ethics, in this view, is not a philosophical afterthought but a core component of clinical excellence.

The ethical dilemmas in end-of-life care extend beyond the moment of death. Wu, Shapiro, and Harrison [24] explore how adverse events and last-minute escalations of care, often made by default rather than design, can erode the moral integrity of hospital teams. They emphasise that ethical failure in such contexts is rarely due to individual malice but rather to institutional inertia, unclear policies, and poor communication. They advocate for a culture of openness, learning, and post-event ethical debriefing, ensuring that moral accountability is distributed rather than concentrated. End-of-life care in hospital settings exposes the fault lines between ethical theory and practice. While autonomy remains a foundational ideal, its practical application is constrained by cognitive decline, institutional inertia, and legal ambiguity. The reviewed literature suggests a paradigm shift: from clinician-centric ethics to system-oriented ethical infrastructures that support shared decision-making, transparent communication, and ethical literacy at every level of care.

The theoretical roots of informed consent are traced by Faden and Beauchamp [44], who define it as both an act of autonomous authorisation and an expression of moral agency. Their work underlines the historical evolution of consent from legal formality to ethical necessity. However, in the digital age, consent has been re-technologised—embedded in interfaces, default checkboxes, and EHR systems that rarely allow for nuanced conversation. This depersonalisation calls for renewed ethical attention, especially when automation mediates the consent process.

Emanuel and Emanuel [49] present a foundational typology of the physician–patient relationship, one that continues to shape disclosure practices. They contrast paternalistic, informative, deliberative, and interpretive models, noting that each implies different responsibilities when medical harm occurs. In the deliberative model, which aligns closely with modern expectations of transparency, physicians are expected to help patients make value-aligned choices, even when outcomes are unfavourable. This model implicitly endorses proactive disclosure, as concealment directly contradicts the principle of mutual deliberation.

Rosenbaum [42] introduces a provocative angle by questioning the moral assumptions of the “less-is-more” movement. While minimalist medicine often aims to reduce iatrogenic harm, it may also become a justification for withholding care in ambiguous situations. Her critique invites ethical scrutiny of cost-containment rhetoric, especially when framed as a safety strategy. The implication is that safety cannot be ethically disentangled from equity; patients deserve not only protection from harm but access to appropriate care, even when uncertain.

### 3.2. Legal Implications

The terminology used here is as follows. Legal responsibility denotes ex ante duties imposed by statute/regulation and case law (e.g., informed-consent jurisprudence; EU AI Act obligations) that hospitals and clinicians must operationalise. Ethical accountability refers to the justificatory standards for decisions beyond mere compliance (autonomy, beneficence, justice), including transparency and reason-giving to patients. Professional liability is the ex post exposure to civil/disciplinary action when duties are breached. Because one may satisfy legal responsibility while falling short ethically (and vice versa), hospitals need governance that aligns these dimensions.

Ethical uncertainty arises not solely from dilemmas of right and wrong but from a lack of clarity regarding professional obligations, informed consent, decision-making authority, and institutional responsibility.

Liability concerns, particularly in high-risk scenarios like robotic surgery or off-label medication use, exert pressure on clinicians and administrators alike [11,12]. Recent literature highlights not only the individual responsibility of clinicians, but also the institutional accountability in ensuring systems-level safety and transparency.

Off-label prescribing offers one of the clearest examples of such uncertainty. Rukundo [1] emphasises how physicians often resort to non-approved therapeutic uses when standard treatment options fail, particularly in palliative contexts or for rare diseases. However, this practice, though legally permissible under certain jurisdictions, lacks consistent oversight mechanisms, leaving clinicians vulnerable to litigation. The ethical conflict lies in balancing beneficence and patient safety while navigating incomplete regulatory coverage. As Rukundo notes, the absence of transparent documentation and patient-specific risk communication undermines the principle of informed consent. Osifowokan and Agbadamasi [2] expand this tension to digital contexts, identifying that AI systems increasingly support or replace clinical reasoning in diagnosis and risk stratification. Their work outlines how legal liability remains unresolved in such scenarios: if an algorithm misleads a physician, is it the coder, the institution, or the clinician who is responsible? In this “shared agency” model, ethical uncertainty is no longer the result of cognitive burden, but institutional silence on where accountability begins or ends [29].

Aldosari and colleagues [48] reinforce this position by detailing the regulatory fragmentation that accompanies the introduction of AI in clinical environments. Their review of Middle Eastern healthcare contexts shows that while national policies may encourage innovation, legal infrastructures lag, leaving gaps in ethical accountability. The authors warn that this misalignment could result in regulatory evasion, where actors exploit policy ambiguity to avoid scrutiny. The long-term consequence, they argue, is a degradation of public trust, particularly in institutions that already struggle with transparency and equitable access. The surgical domain brings a distinct dimension to this discussion. De Paola et al. [26] explore robotic-assisted surgery and its legal implications, arguing that the division of labour between human and machine introduces forensic ambiguity. Who is liable for an error in robotically assisted operations—the surgeon, the software vendor, or the hospital system that approved the platform? This “diffusion of responsibility” undermines not only legal certainty but also ethical self-efficacy: clinicians may defer critical judgment out of deference to machine precision, even when their instincts signal caution.

Rokhshad et al. [30] explore how AI-driven tools used for diagnostics and treatment planning introduce not only technical uncertainty but also ethical variability across institutions. Some clinics use ethically vetted tools with explainable models; others use opaque systems without patient-facing transparency. The inconsistency raises equity concerns, as patients are subject to widely divergent standards of safety depending on the technological maturity of their provider. Kundu and Bardhan [31] bring focus back to regulatory ethics in the Indian context, outlining the lack of unified ethical guidance for AI in neurology. They emphasise that most clinicians operate without clear guidelines, relying instead on informal peer norms or vendor-supplied training. The result is a high level of interpretative discretion, which, while flexible, opens the door to unequal practices and potentially unethical decision-making. At the same time, several authors caution against ascribing moral agency to clinical AI. Sedlakova & Trachsel argue that conversational AI should not be treated as an equal partner in therapeutic dialogue and is best regarded as a tool with restricted functions. Ferdynus further identifies five missing attributes—phenomenal consciousness, intentionality, ethical reflection, prudence, and conscience—that preclude moral agency. In line with these accounts, our analysis treats clinical AI as a tool whose use must remain under human and institutional accountability [32,33].

These themes converge around a central tension: modern technologies have outpaced the ethical and legal systems designed to regulate them. Clinical AI, decision support systems, remote monitoring, and algorithm-based triage all promise to improve care, but they also reassign agency, dilute accountability, and obscure consent. Moreover, ethical literacy among frontline staff remains low, and institutional structures often fail to provide clear policies or escalation pathways when technology-related dilemmas occur. What these studies reveal is a transition point in the ethics of hospital care: from personalist ethics centred on the physician–patient dyad to distributed ethics that consider system designers, data architects, policy-makers, and even non-human agents as co-constituents in clinical decision-making. The call to action is clear: hospitals must develop ethical infrastructures capable of integrating technological advancement without sacrificing patient dignity, autonomy, or safety.

Savioz [31], examining European data privacy frameworks, adds a legal lens to this ethical terrain. The General Data Protection Regulation (GDPR) mandates consent for identifiable data use, yet many healthcare logistics systems operate in a compliance-centred rather than ethics-centred mode.

The ethical and legal risks of algorithmically mediated consent are further emphasised by Roy et al. [7], who point to AI-driven risk scoring and clinical decision support systems that consume and act upon patient data without clinician or patient oversight. When such systems are integrated into EHR platforms, decision pathways become obscured. Physicians may follow recommendations unaware of the data source or the patient’s prior preferences, introducing a subtle erosion of autonomy.

Hospitals must distinguish between access and authorisation, ensuring that family members understand the ethical weight of their role and are not involuntarily implicated in decisions with medico-legal consequences.

In pediatric and adolescent care, Pathak et al. [16] analyse drug disposition practices, revealing how informed consent models fail to account for developmental ethics. Adolescents often occupy a legal gray area—neither full agents nor passive dependents. In pharmacological decisions involving experimental or non-standard dosing, clinicians are left to interpret “assent” within poorly defined frameworks. The authors argue that hospitals must move beyond binary notions of consent and create tiered consent models that respect evolving capacities while upholding clinical rigour. Tkachenko and Pankevych [25] extend the conversation to pharmaceutical ethics and staff responsibilities. In their analysis of Ukrainian hospital pharmacies, they uncover ethical oversights in drug labelling, documentation, and informed interaction with patients. While many systems emphasise pharmacovigilance [34], few incorporate ethics training for frontline pharmaceutical staff. Consent, in such contexts, becomes decoupled from communication; patients are informed but not educated, signed but not engaged.

Dyer [35], focusing on legal developments in patient rights, reinforces the importance of integrating legal transparency into ethical culture. Hospitals must recognise that disclosure policies are not ethical niceties but legal imperatives, grounded in patient autonomy and informed participation. Failing to disclose not only undermines trust but may expose institutions to reputational and legal damage. Ethical silence, in this view, is not neutral—it is actionable. In sum, the reviewed literature reveals a complex interplay between disclosure, institutional design, and ethical climate. Disclosure is not simply an individual virtue but a systemic function, contingent on legal safeguards, organisational transparency, and leadership ethos. As hospitals evolve into technologically dense, multidisciplinary environments, their ethical resilience will depend less on individual heroism and more on collective moral design. Creating environments where ethical speech is safe, expected, and supported is no longer aspirational—it is an operational imperative for ethical and safe care delivery.

### 3.3. Institutional Mechanisms and Governance

The ethics of data use in hospital environments has become increasingly critical with the rise of electronic health records (EHRs), real-time monitoring systems, and the integration of big data for clinical decision-making. Unlike traditional ethical dilemmas centred on direct care relationships, data ethics unfolds at an infrastructural level. Patients often do not interact directly with the systems processing their information, yet their rights, dignity, and autonomy are deeply implicated. This subsection explores how privacy, consent, and ethical accountability are managed—or often mismanaged—in contemporary hospital data systems, with an emphasis on systemic implications.

Stark et al. [10], in a qualitative study based in Germany, reveal the depth of practitioner discomfort regarding the reuse of patient data. While anonymisation protocols are generally respected, clinicians reported unease with secondary data use for research or algorithm development without explicit patient re-consent. Many perceived a mismatch between institutional policies, which emphasise data utility, and their personal ethical intuitions regarding privacy and trust. This disconnect signifies a broader institutional failure to embed ethical reasoning into data workflows, highlighting the need for ethics-aware informatics governance.

Consent is frequently treated as a checkbox, not as an ongoing dialogue. This proceduralism, Savioz argues, strips consent of its normative function and transforms it into an administrative formality. Particularly in vaccine logistics and supply chains, where speed is prioritised, ethical discretion is often subordinated to operational efficiency.

Ref. [9] offers a complementary perspective through his exploration of the ethical burden placed on family caregivers. In many cases, caregivers are granted access to digital health portals and decision-making dashboards without full awareness of the legal implications.

Kass [50], in her public health ethics framework, presents a compelling critique of autonomy-centric models in data governance. She argues that in population-level interventions—such as hospital infection control, quality monitoring, or predictive modelling—collective benefits often override individual rights. While this may be justified in emergencies, it sets a dangerous precedent if uncritically normalised. Kass suggests an ethics framework that balances utility with transparency, ensuring that patients are not only protected but also informed participants in the systems that affect their care.

Roy et al. recommend mandatory transparency protocols, “ethical audit trails,” and human-in-the-loop safeguards to restore normative accountability. In hospital systems where digital literacy is uneven, the burden of understanding these systems often falls on administrators, not patients or clinicians. Gencturk and Erturkmen [36] argue for health ethics infrastructure as a systemic necessity. Their book chapter proposes ethics committees not only for clinical dilemmas, but also for informatics oversight, policy vetting, and data lifecycle governance. Without such mechanisms, hospitals risk reproducing a model where ethics is reactive—applied only when harm is detected, rather than proactively embedded in operational design.

The moral legitimacy of healthcare institutions depends not only on the quality of the treatments they deliver but also on the ethical integrity of their organisational culture. As modern hospitals become increasingly complex systems, ethical dilemmas are no longer limited to bedside interactions—they manifest in how institutions communicate failures, respond to adverse events, and structure accountability. This section examines the interrelation between ethical responsibility, transparency, and institutional safety, building on diverse contributions from bioethics, patient safety science, and policy analysis. Ochieng [37], reflecting from a medical student’s perspective, underscores the formative impact of early exposure to ethical uncertainty in disclosure. His narrative illustrates how clinical trainees often learn not through formal instruction but through observation, where silence about medical errors becomes normalised. This culture of reticence fosters moral disengagement and perpetuates institutional practices that marginalise ethical reflexivity. Ochieng’s account reinforces the call for structured ethics mentoring embedded early in professional formation.

Berlinger, Jennings, and Wolf [46] elaborate on this by advocating for ethics committee engagement beyond consultation, particularly in the development of protocols for adverse event disclosure, informed refusal, and quality oversight. They argue that ethics must be framed not only in terms of reactive response but as a form of preventive architecture, designed to anticipate failures and reduce harm. Their institutional approach repositions ethical competence as a shared responsibility, rather than the moral burden of individual clinicians. Donabedian [43] asserted that the quality of care must be evaluated not only by outcomes but by structures and processes. From this perspective, disclosure of adverse events is not merely a moral act but a quality indicator. Institutions that cultivate cultures of open reporting, ethical learning, and responsive policy demonstrate greater resilience and reduced recurrence of preventable harm. The authors of [29] extend this insight by proposing “resilient healthcare” as a framework for ethical safety. In their view, ethical responsiveness must be designed into systems: in feedback loops, root cause analyses, and team debriefings. Disclosure is reframed not as an act of contrition but as an opportunity for organisational moral learning. Their model integrates ethical awareness with risk management, emphasising that silence about errors constitutes a safety threat in itself.

Wu, Shapiro, and Harrison [24] provide empirical support for this claim. In their multicenter study, they identify a strong correlation between open disclosure policies and staff morale, patient satisfaction, and litigation rates. Importantly, they note that fear of legal consequences, rather than ethical ambivalence, is the principal barrier to transparency. When institutions shield practitioners from punitive fallout, the likelihood of voluntary disclosure increases, reinforcing a feedback loop that benefits all parties. Their findings highlight that ethics and legal strategy are not mutually exclusive but mutually reinforcing when transparency is supported structurally.

Waring and Bishop [38], analysing healthcare policy through a sociological lens, explore how safety initiatives can unintentionally erode ethical trust if implemented without participatory governance. They warn that top-down safety mandates, driven by performance metrics, may create superficial compliance without meaningful engagement. Ethical cultures, they argue, arise from dialogic leadership, not from checklists or fear-based management. In this sense, safety becomes ethical only when it is co-owned by the clinical workforce and institutionally legitimised. Davis et al. [51], in their examination of adverse events in New Zealand hospitals, underscore that most such incidents are not due to gross negligence but to latent system failures—poor communication, unclear documentation, and inadequate supervision. This further complicates the ethics of disclosure: if harm arises from collective oversight rather than individual error, who is responsible for disclosing? Their study suggests that without collective acknowledgment of systemic flaws, the ethical burden will continue to fall unfairly on frontline staff.

Over the past two decades, the ethical orientation of hospital care has expanded beyond individual moral reasoning toward a more systemic understanding of how institutions shape ethical outcomes. In modern healthcare settings, ethics is no longer merely a matter of personal judgment—it has become a property of infrastructure, policy, and organisational logic. This perspective challenges the assumption that moral failures are primarily individual and instead highlights the structural elements that either enable or inhibit ethical performance. Public health frameworks have long emphasised that individual autonomy cannot fully safeguard ethical integrity in complex systems. Instead, proactive institutional design—one that anticipates conflicts and integrates ethical reflexivity—is essential for maintaining moral legitimacy [15]. Such approaches emphasise foresight, equity, and procedural justice, shifting from reactive ethics to ethics by architecture.

This reconceptualisation has significant implications for how hospitals build and sustain ethical cultures. Rather than positioning ethics as the responsibility of committees consulted only in extreme cases, contemporary models suggest it should be diffused throughout organisational processes, from quality assurance to digital transformation [30]. For example, Morgan et al. (2024) document how unit-based ethics rounds embedded within a hospital unit’s routine can normalise ethical discussion and strengthen the unit’s ethical culture, illustrating the benefits of integrating ethics into daily practice [39]. Institutions in such settings tend to demonstrate higher resilience and lower rates of preventable harm—an observation supported by Wehkamp et al., who report measurable improvements in patient safety when ethics is embedded into hospital management systems [40]. A key dimension of this shift is the move toward deliberative ethics structures that enable diverse perspectives to shape institutional decisions, rather than relying solely on top-down mandates. In clinical environments that emphasise person-centredness not only in bedside care but also in administrative policy, both patient satisfaction and clinician morale improve measurably [32]. Moreover, this deliberative infrastructure ensures that institutional norms evolve in step with technological and social change.

From its early formulation as a legal safeguard, consent has transformed into a dynamic moral practice—one that reflects changing power structures, expectations, and patient engagement [33]. Yet this evolution has been uneven across systems. Institutions lacking clear procedural frameworks for consent management often expose both staff and patients to moral distress, particularly when care is fragmented across digital platforms or multiple providers. In clinical reality, decision-making frameworks are only as effective as the systems that support them. When ethics is reduced to codes of conduct or compliance checklists, it risks becoming performative rather than substantive. In contrast, ethically robust organisations cultivate moral literacy through investment in education, structural support for deliberation, and mechanisms for escalating concerns safely. These are not philosophical luxuries; they are critical functions of ethical infrastructure [34].

One of the most significant challenges in institutional ethics lies in recognising that moral failure can be procedural, not only behavioral. For instance, delays in treatment due to bureaucratic bottlenecks or opaque referral pathways can constitute structural harm, even if no clinician has acted maliciously. Ethical design, therefore, must include an assessment of systems risk, where ethical lapses may occur because of the system, not merely within it [28]. This insight has practical implications for policy design. Embedding ethics into patient safety protocols, algorithm audits, and institutional accountability frameworks ensures that organisations do not simply avoid harm but actively foster moral clarity. Whether in decisions about data sharing, triage algorithms, or family engagement, institutions must offer ethically meaningful choices-not just consent forms or disclaimers.

The ethical character of a hospital is not measured by the intensity of its dilemmas but by the maturity of its response systems. As the boundaries between clinical care, technology, and administration continue to blur, ethical resilience must be engineered, not presumed. This entails a deliberate shift from individual moral action to collective moral design-a transformation from being ethics-aware to being ethics-capable.

## 4. Discussion

A salient finding is the tension between patient autonomy and operational pragmatism. While informed consent remains central, in digitally mediated care it often becomes procedural, reduced to checkboxes, delegated surrogacy, or retroactive justification [9,16,17]. This shift mirrors wider concerns about the erosion of agency in data-intensive systems [10,25] and underlines the need for ethical infrastructure, not only ethical literacy [18,33]. Key ethical–legal challenges and the corresponding institutional responses are summarised in Table 4.

Implications for hospital administrators and policy-makers: Prioritise an ethics-by-design operating model. Concretely, tie every digital deployment to a risk register and audit trail; require human-override points for AI-supported decisions; formalise a CIRS–Ethics pathway with escalation to leadership; establish a DPO-co-led data-release committee; and integrate ethics rounds at the unit level to surface value conflicts early. These measures translate abstract principles into repeatable institutional routines that protect patients and staff alike.

A recurring pattern is the institutionalisation of ethical risk in AI and automation: across neurology, dental, and surgical contexts, ethical uncertainty grows when clinical judgment is co-produced by opaque algorithms [7,8,13,14], confirming earlier bioethical warnings about black-box decision support [2,11]. Many hospitals remain underprepared to manage AI-generated harm, lacking legal clarity and operational escalation paths [12,43]. This calls for anticipatory governance that can absorb ethical complexity, not just react to crises, and for stronger disclosure culture: despite the moral imperative, clinicians often lack protection, training, or institutional backing, leaving the burden on individuals rather than systems [26,45,47,48]. In end-of-life care, tensions between autonomy and beneficence surface in human, context-rich decisions [5,15,20]. Frameworks exist [19,24], but their application falters when capacity is compromised or surrogates diverge. The literature supports a shift toward relational, deliberative ethics that prioritises communication and value-clarification over prescriptive checklists [17,46,49].

The review also highlights geopolitical diversity: studies from Uganda, India, Indonesia, and Ukraine show that dilemmas and solutions vary with regulatory maturity and culture [1,4,7,15]. This argues for pluralistic, context-sensitive ethics committees and policies, and for design ethics, embedding ethics proactively in policy, software, infrastructure, and leadership, especially as hospitals adopt hybrid, cross-jurisdictional models of care [28,30,34]. Finally, our synthesis privileges high-quality evidence: core themes and recommendations derive from studies rated high in quality, with medium-quality work used mainly for context, consistent with PRISMA’s emphasis on rigour, enhancing the reliability and relevance of our conclusions.

This review has several limitations. First, although the search strategy covered five major databases and selected grey literature, publication bias and language restrictions (English-only) may have led to under-representation of non-English evidence and negative findings. Second, while classic pre-2015 sources were used solely for conceptual framing, their inclusion may still influence interpretation. Third, most included studies are qualitative, conceptual, or policy-analytic; empirical, interventional evidence remains scarce, which limits causal inference. Fourth, AI-related evidence and regulation are evolving rapidly; regulatory and technical guidance may have changed after our search cut-off (1 April 2025). Finally, heterogeneity across healthcare systems (resources, legal frameworks) constrains generalizability. Future work should include mixed-methods and comparative designs to test the feasibility and impact of the governance measures proposed here.

Future research should empirically test the governance mechanisms outlined here—e.g., human-in-the-loop designs, ethical audit trails, and unit-based ethics rounds—across diverse health-system contexts. Mixed-methods studies linking implementation fidelity to patient-safety and experience outcomes are especially needed, as are comparative evaluations of cross-border telemedicine liability models.

## 5. Conclusions

In light of these findings, hospitals should move from reactive ethical responses to proactive design, integrating human-in-the-loop decision-making, versioned audit trails, and periodic bias/impact assessments into digital deployments; coupling every transformation programme with a legal risk assessment aligned to GMLP and PCCP; clarifying cross-border telemedicine governance (licensure, jurisdiction, disclosure standards); institutionalising ethics rounds and a CIRS–Ethics pathway that routes learning to leadership; establishing a data release process co-led by the DPO; and aligning AI operations with NIST AI RMF 1.0, ISO/IEC 23894, and ISO/IEC 42,001 to ensure continuous monitoring and improvement. In this configuration, ethical resilience becomes a property of systems, not only of individuals.

## Figures and Tables

**Figure 1 healthcare-13-02800-f001:**
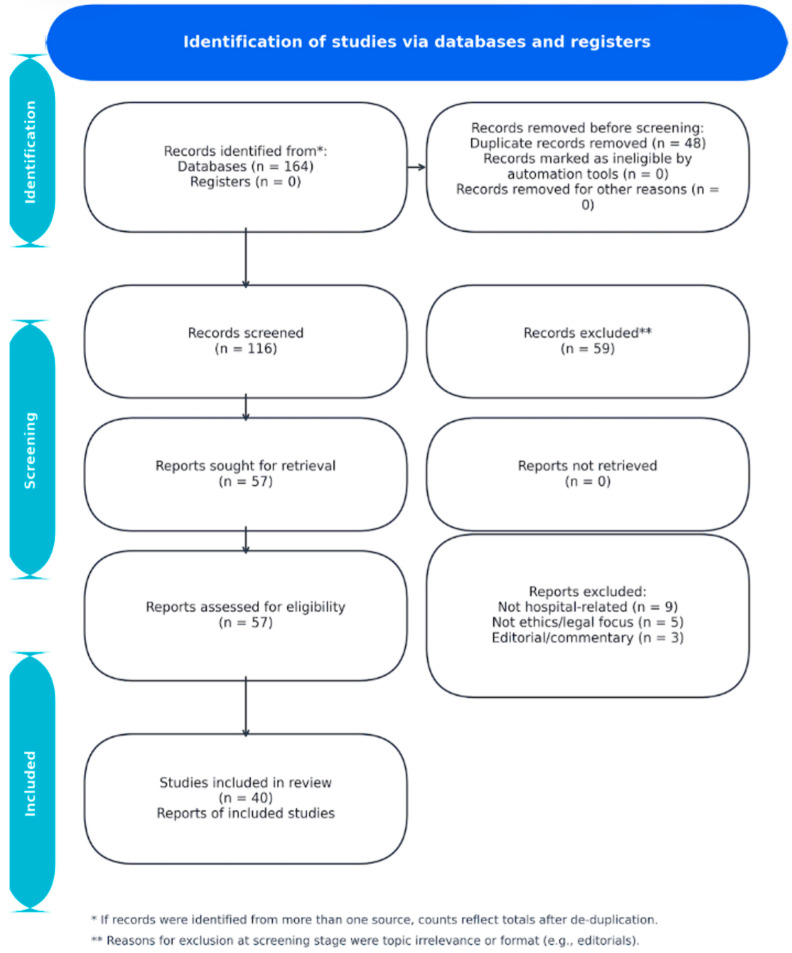
PRISMA flow diagram (records from 2015–2025): 164 identified; 48 duplicates removed; 116 screened; 57 full texts assessed; 40 included.

**Figure 2 healthcare-13-02800-f002:**
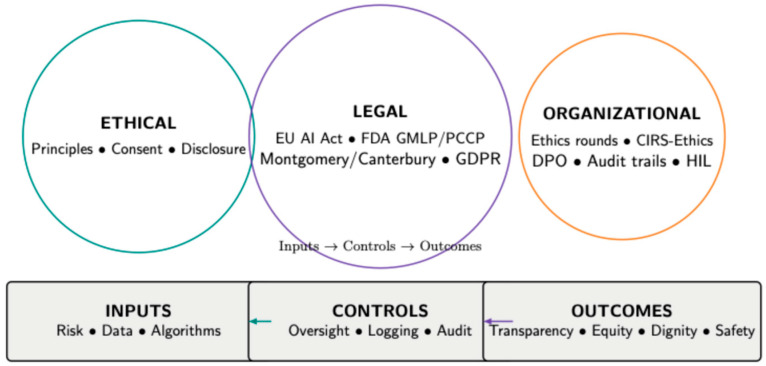
Triadic Governance Model for Ethical Hospital Care. The model aligns three domains—ethical, legal, organizational—and traces the pathway inputs → controls → outcomes. Inputs capture risk, data, and algorithms; controls comprise human oversight, logging, audit, and consent/disclosure routines; outcomes target transparency, equity, dignity, and patient safety. The legend maps each control to its legal anchor (e.g., EU AI Act articles on human oversight and logging; FDA GMLP/PCCP; Montgomery/Canterbury; GDPR) and to a concrete institutional mechanism (e.g., ethics rounds, CIRS-Ethics, and DPO-led data release).

**Table 1 healthcare-13-02800-t001:** Summary of included sources.

Ref. No.	Country	Theme	Context	Study Type	Theoretical Framework	Quality
[1]	Uganda	Off-label prescribing: ethical and legal duties	Pharmacotherapy (steroids)	Ethical/legal analysis	Principlism; professional responsibility	Medium
[21]	China	Innovation networks in advanced medical equipment industry; implications for regional digital-health capacity	Patent co-application network, national vs. YRD comparison; SNA + GeoDetector on drivers	Original research; secondary data; Social Network Analysis + GeoDetector	Innovation-network/core–periphery; SNA metrics (density, centrality, modularity)	High
[22]	China	Effectiveness of new-media pharmacology teaching by background (major/degree)	46 studies, 6447 students; SUCRA ranking	Systematic review + Network meta-analysis	Evidence synthesis with network meta-analysis (SUCRA)	High
[23]	China	Hospital ownership vs. in-hospital mortality and medical expenses	64,171 inpatients across 528 secondary hospitals; PNA/HF/AMI; 2016–2018	Observational, multi-hospital; multilevel logistic/linear regressions	Health-services/ownership–performance lens; multilevel modelling	High
[4]	Indonesia	Legal protection and state responsibility in telemedicine	Telemedicine	Comparative legal analysis	Human rights law; professional liability	Medium
[5]	USA	Ethical issues of PEG in CJD and dementia	Geriatrics/Neurology; enteral feeding	Ethics case commentary	Principlism; proportionality; best interests	High
[6]	Uzbekistan	Ethical and legal issues in simulation-based training	Medical education	Narrative review/policy analysis	Education ethics; competency frameworks	High
[7]	International	AI/ML in healthcare (benefits/risks)	AI in health systems	Review	AI ethics principles	Medium
[8]	USA	AI in acute neurology	Neurology (stroke, ICU)	Editorial	-	High
[9]	-	Legal and ethical challenges for family caregivers	Family caregiving	Dissertation	-	Medium
[10]	Germany	Benefits/risks of health-data reuse for providers	Health-data reuse	Qualitative interview study	-	High
[11]	European Union	Risk-based AI regulation (incl. high-risk health)	AI medical devices and systems	Law (statute)	Risk-based governance	High
[12]	European Union	Operational obligations under the AI Act (high-risk)	AI medical devices and systems	Official overview	Risk taxonomy; conformity assessment	High
[13]	USA/Canada/UK	Good ML Practice for medical device development	AI/ML SaMD	Joint guidance (principles)	GMLP principles	High
[14]	USA	Predetermined Change Control Plan for AI-enabled devices	AI/ML SaMD	Regulatory guidance	PCCP lifecycle management	High
[15]	International	Ethics and governance of AI for health	AI in health (general)	WHO guidance/report	Six core AI ethics principles	High
[16]	International	Ethics and safety of drug disposition in adolescents	Adolescent pharmacology	Book	-	High
[17]	International	Ethics/governance for large multimodal models	LMMs in health	WHO guidance/report	Governance principles for LMMs	High
[18]	UK/France	Patient-safety strategies for the real world	Health systems/patient safety	Book (scholarly)	Safety science (systems; resilience)	High
[19]	United Kingdom	Informed consent standard (material risk; autonomy)	All clinical consent	Case law (UK Supreme Court)	Reasonable patient/material risk	High
[24]	USA	Disclosing adverse events to patients	All clinical settings	Review	CANDOR and communication frameworks	High
[25]	USA	Clinical ethics method (four-box approach)	Clinical decision-making	Book (textbook)	Four-topics method	High
[26]	Italy	Legal/forensic implications in robotic surgery	Robotic surgery	Review/legal analysis	-	High
[27]	Ukraine	Pharmaceutical safety and workforce issues	Pharmacy services	Policy analysis	-	Medium
[28]	USA	Principles of biomedical ethics	All healthcare	Book (theory)	Principlism (four principles)	High
[29]	Saudi Arabia	Challenges of AI in medicine	Health system/AI	Conference proceedings (SHTI)	-	Medium
[30]	International	AI in dentistry: safety, ethics, regulation	Dentistry	Book chapter	Regulatory/ethics principles	High
[31]	India	AI in neurology: ethics, guidelines, law (India)	Neurology/AI	Review/perspective	National guidelines; legal context	High
[32]	Switzerland	Conversational AI in psychotherapy: tool or agent?	Psychotherapy/mental health	Target article/discussion	Agency and responsibility debate	High
[33]	Poland	Why conversational AI is not a moral agent	Psychotherapy/AI	Conceptual analysis	Moral-agency criteria	High
[34]	EU/Switzerland	Navigating EU privacy law	Data protection	Book	GDPR framework	High
[35]	UK	Patient-information duties post-Montgomery	Clinical consent	Legal news/analysis (BMJ)	Reasonable patient/material risk	Medium
[36]	-	Unlocking the Potential of Health Ethics	-	-	-	Medium
[37]	(likely East Africa)	Ethical dilemmas in clinical practice (student perspective)	Clinical training	Viewpoint	-	Medium
[38]	UK	Healthcare quality and safety: policy/practice/research	Health systems	Review	Policy implementation; safety science	High
[39]	USA	Unit-based ethics rounds and ethical culture	Nursing/hospital practice	Qualitative/implementation report	Ethics-rounds model	High
[40]	Germany	Ethical dimensions in CIRS to enhance safety	Incident reporting/patient safety	Qualitative/methodological study	Ethics integration in CIRS	High
[41]	European Union	Human oversight requirement under the AI Act	High-risk AI systems in healthcare (AI medical devices & CDS)	Official explainer / regulatory guidance	Human oversight & deployer obligations (Art. 14)	High

**Table 2 healthcare-13-02800-t002:** Thematic distribution of included studies.

Theme	No. of Studies	Examples
A. Ethical dimensions	19	[5,28,31,42]
B. Legal implications	10	[4,11,13,14,19,26,35]
C. Institutional mechanisms	11	[10,15,17,18,38,39,40,43]

**Table 3 healthcare-13-02800-t003:** Study types represented.

Study Type	Count	Representative Sources
Conceptual/Theoretical (books, theoretical essays, conceptual analyses)	*14*	[28,33]
Review/Analysis (narrative reviews, editorials, book chapters, comparative legal analyses)	*11*	[7,26,38,42]
Policy/Legal/Guidance (statutes, regulations, official guidance, case law)	*8*	[11,13,14,19]
Qualitative (interviews/focus groups; qualitative implementation)	*4*	[10,39,40]
Case study/Case report	*3*	[5]

**Table 4 healthcare-13-02800-t004:** Key Ethical–Legal Challenges and Recommended Institutional Responses.

Challenge	Recommended Institutional Response	Legal/Policy Anchor
Opaque AI decision support	Human-in-the-loop, model cards for clinicians, logging and override procedures	EU AI Act (high-risk), FDA GMLP/PCCP
Secondary data use without meaningful consent	Data-release committee co-led by DPO; layered consent; transparency portal	GDPR; WHO AI Ethics
Cross-border telemedicine liability	Jurisdiction and licensure checklist embedded in telehealth SOPs	National licensure rules; malpractice law
End-of-life decisional conflict	Structured ethics consult; documented value-clarification; unit-based ethics rounds	Hastings Center Guidelines; case law
Off-label/robotic-assisted procedures	Risk–benefit briefing templates; enhanced documentation and consent	Device/medication regulations
Disclosure after harm	CANDOR-style protocols; protected debriefs; feedback to leadership	Disclosure jurisprudence; patient-safety policy

## Data Availability

No new data were created or analysed in this study. Data sharing is not applicable to this article.

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
