# Peer review of "Ethical Dilemmas and Legal Responsibilities in Patient Care: An Analysis of Hospital Safety"

_healthcare, 2025, doi:10.3390/healthcare13212800_

Round 1

Reviewer 1 Report

Comments and Suggestions for Authors

This manuscript undertakes a systematic and conceptually rich exploration of the ethical and legal responsibilities in hospital-based patient care, with emphasis on institutional safety, AI integration, and systemic accountability. The topic is highly relevant, addressing pressing contemporary issues where healthcare ethics intersect with digital transformation, governance, and policy implementation. The paper demonstrates a commendable effort to bridge ethical theory with institutional practice. The comment to revise the manuscript are as follows

  1. The manuscript’s structure is logical but at times conceptually diffuse, with overlapping discussions of ethical theory, organizational risk, and digital governance. Consider restructuring the results and discussion sections to clearly separate ethical dimensions, legal implications, and institutional mechanisms.
  1. Figures or conceptual frameworks summarizing the interrelations between ethical dilemmas, legal accountability, and organizational governance could improve comprehension.
  2. The introduction effectively conveys the problem but needs a stronger theoretical justification for why hospital-based care—as opposed to broader healthcare ecosystems—was selected as the focal setting.
  3. Expand on the literature linking ethical decision-making to organizational resilience, risk management, and patient safety culture.
  4. The review could benefit from integrating comparative perspectives—for instance, differences in ethical and legal frameworks between regions (e.g., EU vs. US or low-resource settings).
  5. While AI and digital governance are discussed, the narrative could be enriched with current regulatory instruments (e.g., EU AI Act, U.S. FDA’s Good Machine Learning Practices, or WHO ethics frameworks for digital health).
  6. Clarify whether the synthesis followed a narrative thematic approach, meta-aggregation, or critical interpretive synthesis.
  7. The manuscript appropriately identifies a shift from individual-level ethical dilemmas to systemic-level vulnerabilities but stops short of explaining why this shift has occurred. Expand the analysis by linking it to digitization, bureaucratization, and the growing use of predictive technologies.
  8. The discussion on AI ethics should include deeper analysis of algorithmic bias, data privacy challenges, and accountability gaps—particularly who bears responsibility when harm results from algorithmic decisions.
  9. Integrate mechanistic explanations: how exactly do poor governance structures or unclear accountability pathways generate ethical risks in hospital systems?
  10. “Ethics-by-design” and “organizational reflexivity” are key contributions but should be grounded in practical exemplars—such as hospital ethics committees, digital audit trails, or data ethics oversight bodies.
  11. While ethical reasoning is strong, the legal dimension remains underdeveloped. The paper should engage more substantively with: Case laws or precedents illustrating hospital liability or patient rights violations.
    1. decision systems.
  12. Distinguishing between legal responsibility, ethical accountability, and professional liability would help clarify the manuscript’s conceptual boundaries.
  13. The concluding section should translate findings into specific policy recommendations, such as: Development of institutional “ethics-by-design” protocols, integration of legal risk assessments into digital transformation plans and Strengthening cross-border governance mechanisms for telemedicine.
  14. Highlight how healthcare administrators, regulatory bodies, and ethics committees can operationalize these recommendations.
  15. Consider including a framework or model for ethical governance in hospitals, synthesizing your findings into a visual or conceptual schema.
  1. The following recent studies are suggested to evaluate and add to the introductory literature the manuscript: https://doi.org/10.3389/fpubh.2025.1635475, https://doi.org/10.1016/j.ejphar.2025.177255, https://doi.org/10.1186/s13690-023-01029-y

Reviewer 2 Report

Comments and Suggestions for Authors

The article aligns with the scope of the journal and addresses an important topic. Nevertheless, in some places, certain additions, clarifications, or corrections are needed, as outlined below:

Lines 198–199: It is true that the question of who is responsible for an action becomes crucial when many parties are involved, including AI. There is already a body of literature that addresses this issue, and it would be worthwhile to at least indicate how such problems can be resolved or are being resolved.

Lines 248 and 233: The authors use the phrases “ethical opacity” and, at other times, “moral opacity.” These terms should be unified. The term "morality" ("moral") should not be used interchangeably with "ethics" ("ethical") as – unless clearly defined – they can mean entirely different things. Ethics provides justifications – there are various ethical theories for this purpose – while morality often describes human experience. It would be better to consistently use the term “ethical opacity” throughout the article.

Line 273: It is unclear whether the use of such a semantically complex term as “personhood” is justified in this context. The term should either be explained here or replaced with a more neutral one to avoid ambiguity.

Lines 319–321: The literature already contains numerous arguments showing that AI cannot be treated as an agent (actor), but only as a tool (See, for example: Sedlakova, J., & Trachsel, M. (2022). Conversational Artificial Intelligence in Psychotherapy: A New Therapeutic Tool or Agent? The American Journal of Bioethics, 23(5), 4-13; Ferdynus, M. P. (2023). Five reasons why a conversational artificial intelligence cannot be treated as a moral agent in psychotherapy. Archives of Psychiatry and Psychotherapy, 25(4), 26-29). I believe this point should be added, as the opinion expressed in the article may give the impression that it is the only or universally accepted view.

Line 400: Ethics is a philosophical discipline. Attitudes can be morally acceptable or not; behavior can be ethical or unethical – aligned or misaligned with medical/nursing codes of ethics, etc. However, ethics is always a philosophical theory. This sentence must be revised, as its current form suggests a misunderstanding of what ethics fundamentally is.

Line 468: The text cites “Kass [25]”, but it should be “Kass [36]”. This is a reference error.

General Comment: The article lacks a section discussing its limitations. It would be beneficial to add a few sentences addressing this issue before the conclusion.

Reviewer 3 Report

Comments and Suggestions for Authors

This is a well-written and thoughtfully structured review. The purpose is clearly articulated, and the synthesis of ethical and legal perspectives in hospital safety is both comprehensive and insightful. However, I have some concerns regarding the inclusion criteria and quality selection of the reviewed studies. Although the authors stated that only articles published between 2015 and 2025 were included, several seminal but older works such as Donabedian (1966) were also part of the dataset. While these classical references are valuable for theoretical framing, their inclusion within the analytical pool may compromise the temporal consistency of the review.

Additionally, the 40 selected studies comprise a mix of medium- and high-quality works. To strengthen the rigor and impact of the findings, I would suggest focusing the synthesis and discussion primarily on the high-quality studies, while referencing the medium-quality ones only for contextual support. This selective approach would enhance the credibility of the conclusions and align the review more closely with PRISMA’s emphasis on methodological robustness.

I would be glad to review the revised version once the authors have addressed these points.

Round 2

Reviewer 1 Report

Comments and Suggestions for Authors

The authors have adequately addressed all reviewer comments, and the revised manuscript is now clear, comprehensive, and suitable for publication in its current form.

Author Response

We sincerely thank the reviewer for the positive evaluation and kind recommendation for publication. We greatly appreciate the time and effort dedicated to reviewing our work.

Reviewer 2 Report

Comments and Suggestions for Authors

I have no more comments.

Author Response

We thank the reviewer for the previous feedback and for confirming that no further comments are required. We greatly appreciate the time and attention dedicated to evaluating our manuscript.

Reviewer 3 Report

Comments and Suggestions for Authors

Thank you for providing me with the revised version of the manuscript. It demonstrates substantial improvement. However, there are a few minor concerns need authors’ attention:

  • Ensure consistent referencing style and correct citation numbering (some brackets and numbering [e.g., “24,25”] appear inconsistent in a few places).
  • Review long sentences (e.g., in the Introduction and Discussion) for readability and flow; breaking them into shorter, clearer statements could improve accessibility for non-native readers.
  • The Triadic Governance Model (Figure 2) is a strong visual, but it may benefit from a brief caption explanation of “Inputs → Controls → Outcomes” to ensure standalone clarity.
  • Consider adding a concise summary figure or table in the Discussion showing “Key Ethical-Legal Challenges and Recommended Institutional Responses.”
  • The Discussion could more explicitly highlight implications for hospital administrators or policy-makers, translating theoretical insights into actionable points.
  • Add a brief future research direction paragraph (e.g., need for empirical validation of governance models in diverse healthcare systems).
  • Ensure uniform terminology (e.g., “ethical-legal” vs. “legal-ethical,” “AI” vs. “artificial intelligence”) for professional consistency.
  • Check alignment of subheadings (3.1, 3.2, 3.3) and ensure consistent formatting according to the journal’s style.
